# Finite-Dimensional BFRY Priors and Variational Bayesian Inference for Power Law Models

**Juho Lee**
POSTECH, Korea
stonecold@postech.ac.kr

**Lancelot F. James**
HKUST, Hong Kong
lancelot@ust.hk

**Seungjin Choi**
POSTECH, Korea
seungjin@postech.ac.kr

## Abstract

Bayesian nonparametric methods based on the Dirichlet Process (DP), gamma process and beta process, have proven effective in capturing aspects of various datasets arising in machine learning. However, it is now recognized that such processes have their limitations in terms of the ability to capture power law behavior. As such there is now considerable interest in models based on the Stable Processs (SP), Generalized Gamma process (GGP) and Stable-Beta Process (SBP). These models present new challenges in terms of practical statistical implementation. In analogy to tractable processes such as the finite-dimensional Dirichlet process, we describe a class of random processes, we call iid finite-dimensional BFRY processes, that enables one to begin to develop efficient posterior inference algorithms such as variational Bayes that readily scale to massive datasets. For illustrative purposes, we describe a simple variational Bayes algorithm for normalized SP mixture models, and demonstrate its usefulness with experiments on synthetic and real-world datasets.

## 1   Introduction

Bayesian non-parametric ideas have played a major role in various intricate applications in statistics and machine learning. The Dirichlet process (DP) [1], due to its remarkable properties and relative tractability, has been the primary choice for many applications. It has also inspired the development of variants such as the HDP [2] which can be seen as an infinite-dimensional extension of latent Dirichlet allocation [3]. While there are many possible descriptions of a DP, a most intuitive one is its view as the limit, as $K \to \infty$, of a finite-dimensional Dirichlet process, $P_K(A) = \sum_{k=1}^{K} D_k \mathbb{I}_{\{V_k \in A\}}$, where one can take $(D_1, \ldots, D_K)$ to be a $K$-variate symmetric Dirichlet vector on the $(K-1)$-simplex with parameters $(\theta/K, \ldots, \theta/K)$, for $\theta > 0$ and $\{V_k\}$ are an arbitrary i.i.d. sequence of variables over a space $\Omega$, with law $H(A) = \Pr(V_k \in A)$. Multiplying by a $G_\theta$, an independent $\text{Gamma}(\theta, 1)$, variable, leads to a finite-dimensional Gamma process $\Gamma_K(A) = \sum_{k=1}^{K} G_k \mathbb{I}_{\{V_k \in A\}} := G_\theta P_K(A)$, where $\{G_k\}$ are i.i.d. $\text{Gamma}(\theta/K, 1)$ variables, and one may set $\Gamma_K(\Omega) = G_\theta$. It was shown that $\lim_{K \to \infty}(P_K, \Gamma_K) \stackrel{d}{=} (\tilde{F}_{0,\theta}, \tilde{\mu}_{0,\theta})$, where the limits correspond to a DP and a Gamma process (GP) [4]. While $(P_K, \Gamma_K)$ are often viewed as approximations to the DP and Gamma process (GP), the works of [5, 6, 7] and references therein demonstrate the general utility of these models.

The relationship between the GP and DP shows that the GP is a more flexible random process. This is borne out by its recognized applicability for a wider range of data structures. As such, it suffices to focus on $\Gamma_K$ as a tractable instance of what we refer to as an i.i.d. finite-dimensional process. In general, we say a random process, $\mu_K(\cdot) := \sum_{k=1}^{K} J_k \delta_{V_k}$, is an *i.i.d. finite-dimensional process* if [(i)] For each fixed $K$, $(J_1, \ldots, J_K)$ are i.i.d. random variables [(ii)] $\lim_{K \to \infty} \mu_K \stackrel{d}{=} \tilde{\mu}$, where $\tilde{\mu}$ is a completely random measure (CRM) [8]. In fact, from [9] (Theorem 14), it follows that if the limit exists $\tilde{\mu}$ must be a CRM and therefore $T := \tilde{\mu}(\Omega) < \infty$ is a non-negative infinitely divisible random

variable. On the other hand, it is important to note that, $\{J_k\}$ and $T_K = \mu_K(\Omega) = \sum_{k=1}^K J_k$ need not be infinitely divisible. We also point out there are many constructions of $\mu_K$ that converge to the same $\tilde{\mu}$. According to [4], for every CRM $\tilde{\mu}$ one can construct special cases of $\mu_K$ that always converge as follow: Let $(C_1, \ldots, C_K)$ denote a disjoint partition of $\Omega$ such that $H(C_k) = 1/K$ for $k = 1, \ldots, K$, then one can set $J_k \stackrel{d}{=} \tilde{\mu}(C_k)$, where the $\{J_k\}$ are iid infinitely divisible variables and $T_K \stackrel{d}{=} T$. For reference we shall call such $\mu_K$ finite-dimensional Kingman processes or simply *Kingman proceses*. It is clear that the finite-dimensional gamma process satisfies such a construction with $J_k = G_k$ and $T_K = G_\theta$. However, the nice tractable features of this special case, do not carry over in general. This is due to the fact that there are many cases where the distribution of $\tilde{\mu}(C_k)$, is not tractable either in the sense of not being easily simulated or having a relatively simple density. The latter is of particular importance if one wants to consider developing inferential techniques for CRM models that scale readily to large or massive data sets. An example of this would be variational Bayes type methods, which would otherwise be well suited to the i.i.d. based models [10]. As such we consider a finer class of i.i.d. finite-dimensional processes as follows: We say $\mu_K$ is *ideal* if in addition to [(i)] and [(ii)] it satisfies [(iii)] the $J_k$ are easily simulated [(iv)] the density of $J_k$ has an explicit closed form suitable for application of techniques such as variational Bayes. We do not attempt to specify any formal structure on what we mean by *ideal*, except to note that one can easily recognize a choice of $\mu_K$ that is not *ideal*.

Our focus in this paper is not to explore the generalities of finite-dimensional processes. Rather it is to identify specific *ideal* processes which are suitable for important cases where $\tilde{\mu}$ is a Stable process (SP), or Generalized Gamma process (GGP). Furthermore by a simple transformation we can construct processes that have behaviour similar to a Stable-Beta process (SBP). The SP, GGP, SBP, and processes constructed from them, are now regarded as important alternatives to the DP, GP and beta process (BP), as they, unlike the (DP, GP, BP), are better able to capture power law behavior inherent in many types of datasets [11, 12, 13, 14, 15]. Unfortunately Kingman processes based on SP, GGP or SBP are clearly not *ideal*. Indeed, if one considers for $0 < \alpha < 1$, $T = S_\alpha$ a positive stable random variable, with density $f_\alpha$, then the corresponding stable process $\tilde{\mu}_{\alpha,0}$, is such that $J_k \stackrel{d}{=} \mu_{\alpha,0}(C_k) \stackrel{d}{=} K^{-1/\alpha} S_\alpha$. While it is fairly easy to sample $S_\alpha$ and hence each $J_k$, it is well-known that the density $f_\alpha$ does not have generally a tractable form. Things become worse in the GGP setting as the relevant density is formed by exponentially tilting the density $f_\alpha$. Finally it is neither clear from the literature how to sample $T$ for SBP, and much less have a simple form for its corresponding density. Here we shall construct ideal processes based on various manipulations of a class of $\mu_K$ we call finite-dimensional BFRY [16] processes. We note that BFRY random variables appear in machine learning contexts in recent work [17], albeit in a very different role.

Based on finite-dimensional BFRY processes, we provide simple variational Bayes algorithms for mixture models based on normalized SP and GGP. We also derive collapsed variational Bayes algorithms where the jumps are marginalized out. We demonstrate the effectiveness of our approach on both synthetic and real-world datasets. Our intent here is to demonstrate how these processes can be used within the context of variational inference. This in turn hopefully helps to elucidate how to implement such procedures, or other inference techniques that benefit from explicit densities, such as hybrid Monte Carlo [18] or stochastic gradient MCMC algorithms [19].

## 2 Background

### 2.1 Completely random measure and Laplace functionals

Let $(\Omega, \mathcal{F})$ be a measurable space, A random measure $\mu$ on $\Omega$ is completely random [8] if for any disjoint $A, B \in \mathcal{F}$, $\mu(A)$ and $\mu(B)$ are independent. It is known that any CRM can be written as the sum of a deterministic measure, a measure with fixed atoms, and a random measure represented as a linear functional of the Poisson process [8]. In this paper, we focus on CRMs with only the third component. Let $\Pi$ be a Poisson process on $\mathbb{R}^+ \times \Omega$ with mean intensity decomposed as $\nu(ds, d\omega) = \rho(ds)H(d\omega)$. A realization of $\Pi$ and corresponding CRM is written as

$$\Pi = \sum_{k=1}^{\Pi(\mathbb{R}^+, \Omega)} \delta_{(s_k, \omega_k)}, \quad \mu = \int_0^\infty s\Pi(ds, d\omega) = \sum_{k=1}^{\Pi(\mathbb{R}^+, \Omega)} s_k \delta_{\omega_k}. \tag{1}$$

We refer to $\rho$ as the *Lévy measure* of $\mu$ and $H$ as the *base measure*, and write $\mu \sim \mathrm{CRM}(\rho, H)$. Examples of CRMs include the gamma process $\mathrm{GP}(\theta, H)$ with Lévy measure $\rho(ds) = \theta s^{-1} e^{-s} ds$ or the beta process $\mathrm{BP}(c, \theta, H)$ with Lévy measure $\rho(du) = \theta c u^{-1}(1-u)^{c-1}\mathbb{I}_{\{0 \leq u \leq 1\}} du$. Stable, generalized gamma, and stable beta are also CRMs, and we will discuss them later.

A CRM is identified by its *Laplace functional*, just as a random variable is identified by its characteristic function [20]. For a random measure $\mu$ and a measurable function $f$, the Laplace functional $\mathcal{L}_\mu(f)$ is defined as

$$\mathcal{L}_\mu(f) := \mathbb{E}[e^{-\mu(f)}], \quad \mu(f) := \int_\Theta f(\omega)\mu(d\omega). \tag{2}$$

When $\mu \sim \mathrm{CRM}(\rho, H)$, the Laplace functional can be computed using the following theorem.

**Theorem 1.** *(Lévy-Khintchine Formula [21]) For $\mu \sim \mathrm{CRM}(\rho, H)$ and measurable $f$ on $\Omega$,*

$$\mathcal{L}_\mu(f) = \exp\left\{ -\int_\Omega \int_0^\infty (1 - e^{-sf(\omega)})\rho(ds)H(d\omega) \right\}. \tag{3}$$

### 2.2 Stable and related processes

A Stable Process $\mathrm{SP}(\theta, \alpha, H)$ is a CRM with Lévy measure

$$\rho(ds) = \frac{\theta}{\Gamma(1-\alpha)} s^{-\alpha-1} ds, \tag{4}$$

and a Generalized Gamma Process $\mathrm{GGP}(\theta, \alpha, \tau, H)$ is a CRM with Lévy measure

$$\rho(ds) = \frac{\theta}{\Gamma(1-\alpha)} s^{-\alpha-1} e^{-\tau s} ds, \tag{5}$$

where $\theta > 0$, $0 < \alpha < 1$, and $\tau > 0$.

GGP is general in the sense that we can get many other processes from it. For example, by letting $\alpha \to 0$ we get GP, and by setting $\tau = 0$ we get SP. Furthermore, while it is well-known that the Pitman-Yor process (see [22] and [23]) can be derived from SP, there is also a construction based on GGP as follows. In particular as a consequence of ([23], Proposition 21), if we randomize $\theta = \mathrm{Gamma}(\theta'/\alpha, 1)$ in SP and normalize the jumps, then we get the Pitman-Yor process $\mathrm{PYP}(\theta', \alpha)$ for $\theta' > 0$. The jumps of SP and GGP are known to be heavy-tailed, and this results in power-law behaviour of data drawn from models having those processes as priors.

The stable beta process $\mathrm{SBP}(\theta, \alpha, c, H)$ is a CRM with Lévy measure

$$\rho(du) = \frac{\theta\Gamma(1+c)}{\Gamma(1-\alpha)\Gamma(c+\alpha)} u^{-\alpha-1}(1-u)^{c+\alpha-1}\mathbb{I}_{\{0 \leq u \leq 1\}} du, \tag{6}$$

where $\theta > 0$, $0 < \alpha < 1$, and $c > -\alpha$. SBP can be viewed as a heavy-tailed extension of BP, and the special case of $c = 0$ can be obtained by applying the transformation $u = s/(s+1)$ in SP.

### 2.3 BFRY distributions

The BFRY distribution with parameter $0 < \alpha < 1$, written as $\mathrm{BFRY}(\alpha)$, is a random variable with density

$$g_\alpha(s) = \frac{\alpha}{\Gamma(1-\alpha)} s^{-\alpha-1}(1 - e^{-s}). \tag{7}$$

We can simulate $S \sim \mathrm{BFRY}(\alpha)$ with $S \stackrel{d}{=} G/B$, where $G \sim \mathrm{Gamma}(1-\alpha, 1)$ and $B \sim \mathrm{Beta}(\alpha, 1)$. One can easily verify this by computing the density of the ratio distribution.

The name BFRY was coined in [16] after the work of Bertoin, Fujita, Roynette, and Yor [24] who obtained explicit descriptions of the infinitely divisible random variable and subordinator corresponding to the density. However, the density arises much earlier, and can be found in a variety of contexts, for instance in [23] (Proposition 12, Corollary 13 and see also Eq.(124)) and [25]. See

[17] for the use of BFRY distributions to induce the closed form Indian buffet process type generative processes that have a type III power law behaviour.

We also explain some variations of BFRY distributions needed for the construction of finite-dimensional BFRY processes for SP and GGP. First, we can scale the BFRY random variables by some scale $c > 0$. In that case, we write $S \sim \mathrm{BFRY}(c, \alpha)$, and the density is given as

$$g_{c,\alpha}(s) = \frac{c}{\Gamma(1-\alpha)} s^{-\alpha-1}(1 - e^{-(\alpha/c)^{1/\alpha}s}). \tag{8}$$

We can easily sample $S \sim \mathrm{BFRY}(c, \alpha)$ as $S \overset{d}{=} (\alpha/c)^{-1/\alpha}T$ where $T \sim \mathrm{BFRY}(\alpha)$. We can also exponentially tilt the scaled BFRY random variable, with a parameter $\tau > 0$. For that we write $S \sim \mathrm{BFRY}(c, \tau, \alpha)$, and the density is given as

$$g_{c,\tau,\alpha}(s) = \frac{\alpha s^{-\alpha-1}e^{-\tau s}(1 - e^{-(\alpha/c)^{1/\alpha}s})}{\Gamma(1-\alpha)\{(\tau + (\alpha/c)^{1/\alpha})^{\alpha} - \tau^{\alpha}\}}. \tag{9}$$

We can simulate $S \sim \mathrm{BFRY}(c, \tau, \alpha)$ as $S \overset{d}{=} GT$ where $G \sim \mathrm{Gamma}(1 - \alpha, 1)$ and $T$ is a random variable with density,

$$h(t) = \frac{\alpha t^{-\alpha-1}}{(\tau + (\alpha/c)^{1/\alpha})^{\alpha} - \tau^{\alpha}} \mathbb{I}_{\{(\tau+(\alpha/c)^{1/\alpha})^{-1} \leq t \leq \tau^{-1}\}}, \tag{10}$$

which can easily be sampled using inverse transform sampling.

## 3  Main Contributions

### 3.1  A Motivating example

Before we jump into our method, we first revisit an example of ideal finite-dimensional processes. Inspired by constructions of DP and GP, the Indian buffet process (IBP, [26]) was developed as a model for feature selection, by considering the limit $K \to \infty$ of an $M \times K$ binary matrix whose entries $\{Z_{m,k}\}$ are conditionally independent $\mathrm{Bern}(U_k)$ variables where $\{U_k\}$ are i.i.d. $\mathrm{Beta}(\theta/K, 1)$ variables. Although not explicitly described as such, this leads to the notion of a finite-dimensional beta process $\mu_K = \sum_{k=1}^{K} U_k \delta_{V_k}$. In [26], IBP was obtained as the limit of the marginal distribution where $\mu_K$ was marginalized out, and this result coupled with [27] show indirectly that $\lim_{K \to \infty} \mu_K \to \mu \sim \mathrm{BP}(\theta, H)$. Here, we show another proof of this convergence, by inspecting the Laplace functional of $\mu_K$. The Laplace functional of $\mu_K$ is computed as follows:

$$
\begin{aligned}
\mathcal{L}_{\mu_K}(f) &= \mathbb{E}[e^{-\mu_K(f)}] = \left[ \int_\Omega \int_0^1 \frac{\theta}{K} u^{\frac{\theta}{K}-1} e^{-uf(\omega)} du\, H(d\omega) \right]^K \\
&= \left[ 1 - \frac{1}{K} \int_\Omega \int_0^1 \theta u^{\frac{\theta}{K}-1}(1 - e^{-uf(\omega)}) du\, H(d\omega) \right]^K.
\end{aligned}
\tag{11}
$$

Since $u^{\theta/K}$ is bounded by 1, the bounded convergence theorem implies

$$\lim_{K \to \infty} \mathcal{L}_{\mu_K}(f) = \exp\left\{ -\int_\Omega \int_0^\infty (1 - e^{-uf(\omega)})\theta u^{-1}\mathbb{I}_{\{0 \leq u \leq 1\}} du\, H(d\omega) \right\}, \tag{12}$$

which exactly matches the Laplace functional of $\mu$ computed by Eq. (3). In contrast to the marginal likelihood arguments, in our proof, we illustrate the direct relationship between the random measures and suggest a blueprint that can be applied to other CRMs. Note that the finite-dimensional beta process is not a Kingman process, since the beta variables are not infinitely divisible and the total mass $T$ is a Dickman variable. We can also apply our argument to the case of the finite-dimensional gamma process, the proof of which is given in our supplementary material.

### 3.2  Finite-dimensional BFRY processes

Inspired by the finite-dimensional beta and gamma process examples, we propose *finite-dimensional BFRY processes*, which converge to SP, GGP, and SBP as $K \to \infty$.

**Theorem 2.** *(Finite-dimensional BFRY processes)*

*(i) Let $\mu \sim \mathrm{SP}(\theta, \alpha, H)$. Construct $\mu_K$ as follows:*

$$J_1, \ldots, J_K \overset{\text{i.i.d.}}{\sim} \mathrm{BFRY}(\theta/K, \alpha), \quad V_1, \ldots, V_K \overset{\text{i.i.d.}}{\sim} H, \quad \mu_K = \sum_{k=1}^{K} J_k \delta_{V_k}. \tag{13}$$

*(ii) Let $\mu \sim \mathrm{GGP}(\theta, \alpha, \tau, H)$. Construct $\mu_K$ as follows:*

$$J_1, \ldots, J_K \overset{\text{i.i.d.}}{\sim} \mathrm{BFRY}(\theta/K, \tau, \alpha), \quad V_1, \ldots, V_K \overset{\text{i.i.d.}}{\sim} H, \quad \mu_K = \sum_{k=1}^{K} J_k \delta_{V_k}. \tag{14}$$

*(iii) Let $\mu \sim \mathrm{SBP}(\theta, \alpha, 0, H)$. Construct $\mu_K$ as follows:*

$$S_1, \ldots, S_K \overset{\text{i.i.d.}}{\sim} \mathrm{BFRY}(\theta/K, \alpha), \quad J_k = \frac{S_k}{S_k + 1} \quad \text{for } k = 1, \ldots, K,$$

$$V_1, \ldots, V_K \overset{\text{i.i.d.}}{\sim} H, \quad \mu_K = \sum_{k=1}^{K} J_k \delta_{V_k}. \tag{15}$$

*For all three cases, $\lim_{K \to \infty} \mathcal{L}_f(\mu_K) = \mathcal{L}_f(\mu)$ for an arbitrary measurable $f$.*

*Proof.* We first provide a proof for SP case (i), and the proof for GGP (ii) is almost identical. The Laplace functional of $\mu_K$ is written as

$$
\begin{aligned}
\mathcal{L}_{\mu_K}(f) &= \left[ \int_\Omega \int_0^\infty e^{-sf(\omega)} \frac{\theta}{K\Gamma(1-\alpha)} s^{-\alpha-1} (1 - e^{-(\alpha K/\theta)^{1/\alpha} s}) ds H(d\omega) \right]^K \\
&= \left[ 1 - \frac{1}{K} \int_\Omega \int_0^\infty \frac{\theta}{\Gamma(1-\alpha)} (1 - e^{-sf(\omega)}) s^{-\alpha-1} (1 - e^{-(\alpha K/\theta)^{1/\alpha} s}) ds H(d\omega) \right]^K
\end{aligned}
$$

Since $1 - e^{-(\alpha K/\theta)s}$ is bounded by 1, the bounded convergence theorem implies,

$$\lim_{K \to \infty} \mathcal{L}_{\mu_K}(f) = \exp \left\{ - \int_\Omega \int_0^\infty (1 - e^{-sf(\omega)}) \frac{\theta}{\Gamma(1-\alpha)} s^{-\alpha-1} ds H(d\omega) \right\},$$

which exactly matches the Laplace functional of SP. The proof of (iii) is trivial from (i) and the relationship between SP and SBP. $\square$

**Corollary 1.** *Let $\tau = 1$ and $\alpha \to 0$ in (14). Then $\mu_K$ will converge to $\mu \sim \mathrm{GP}(\theta, H)$.*

*Proof.* The result is trivial by letting $\alpha \to 0$ in $\mathcal{L}_f(\mu_K)$. $\square$

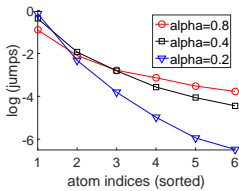

Figure 1: Log of average jump sizes of NSPs

Finite-dimensional BFRY processes are certainly ideal processes, since we can easily sample the jumps $\{J_k\}$, and we have explicit closed form densities written as (8) and (9). Hence, based on those processes, we can develop efficient inference algorithms such as variational Bayes for power-law models related to SP, GGP, and SBP that require explicit densities of jumps. Figure 1 illustrates the log of average jump sizes of 100 normalized SPs drawn using finite-dimensional BFRY processes, with $\theta = 1$, $K = 1000$, and varying $\alpha$. As expected, the jumps generated with bigger $\alpha$ are more heavy-tailed.

## 3.3 Finite-dimensional normalized random measure mixture models

A normalized random measure (NRM) is obtained by normalizing a CRM by its total mass. A NRM mixture model (NRMM) is then defined as a mixture model with NRM prior, and its generative process is written as follows:

$$\mu \sim \mathrm{CRM}(\rho, H), \quad \phi_1, \ldots \phi_N \overset{\text{i.i.d.}}{\sim} \mu/\mu(\Omega), \quad X_n | \phi_n \sim L(\phi_n), \tag{16}$$

where $L$ is a likelihood distribution. One can easily do posterior inferences by marginalizing out $\mu$, with an auxiliary variable. Once $\mu$ is marginalized out we can develop a Gibbs sampler [28]. However, this scales poorly as mentioned earlier. On the other hand, one may replace $\mu$ with $\mu_K$, yielding the finite-dimensional NRMM (FNRMM), for which efficient variational Bayes can be developed provided that the finite-dimensional process is ideal.

### 3.4 Variational Bayes for finite-dimensional mixture models

We first introduce a variational Bayes algorithm for finite-dimensional normalized SP mixture (FNSPM). The joint likelihood of the model is written as

$$\Pr(\{X_n \in dx_n, z_n\}, \{J_k \in ds_k, V_k \in d\omega_k\}) = s_{\cdot}^{-N} \prod_{k=1}^{K} s_k^{N_k} g_{\theta/K,\alpha}(ds_k) \prod_{z_n=k} L(dx_n|\omega_k) H(d\omega_k), \quad (17)$$

where $s_{\cdot} := \sum_k s_k$, and $z_n$ is an indicator variable such that $z_n = k$ if $\phi_n = \omega_k$. We found it convenient to introduce an auxiliary variable $U \sim \mathrm{Gamma}(N, s_{\cdot})$ as in [20] to remove $s_{\cdot}^{-N}$:

$$\Pr(\{X_n \in dx_n, z_n\}, \{J_k \in ds_k, V_k \in d\omega_k\}, U \in du)$$

$$\propto \quad u^{N-1} du \prod_{k=1}^{K} s_k^{N_k} e^{-us_k} g_{\theta/K,\alpha}(s_k) ds_k \prod_{z_n=k} L(dx_n|\omega_k) H(d\omega_k). \quad (18)$$

Now we introduce variational distributions for $\{z, s, \omega, u\}$ and optimize the Evidence Lower BOund (ELBO) with respect to the parameters of the variational distributions. The posterior statistics can be simulated using the optimized variational distributions. We can also optimize the hyperparamters $\theta$ and $\alpha$ with ELBO. The detailed optimization procedure is described in the supplementary material.

### 3.5 Collapsed Gibbs sampler for finite-dimensional mixture models

As with the NRMM, we can also marginalize out the jumps $\{J_k\}$ to get the collapsed model. Marginalizing out $s$ in (18) gives

$$\Pr(\{X_n \in dx_n, z_n\}, \{V_k \in d\omega_k\}, U \in du) \propto u^{N-1} du \prod_{k=1}^{K} \left[ \frac{\theta(1 - \xi^{N_k - \alpha})}{u^{N_k - \alpha}} \frac{\Gamma(N_k - \alpha)}{\Gamma(1 - \alpha)} \right]^{\mathbb{I}_{\{N_k > 0\}}}$$

$$\times \left[ \frac{\theta u^{\alpha}}{\alpha} (\xi^{-\alpha} - 1) \right]^{\mathbb{I}_{\{N_k = 0\}}} \prod_{z_n=k} L(dx_n|\omega_k) H(d\omega_k), \quad (19)$$

where $\xi := \frac{u}{u + (\alpha K/\theta)^{1/\alpha}}$. Based on this, we can derive the collapsed Gibbs sampler for FNSPM, and the detailed equations are in the supplementary material.

### 3.6 Collapsed variational Bayes for finite-dimensional mixture models

Based on the marginalized log likelihood (19), we can develop a collapsed variational Bayes algorithm for FNSPM, following the collapsed variational inference algorithm for DPM [29]. We introduce variational distributions for $\{u, z, \omega\}$, and then the update equation for $q(z)$ is computed using the conditional posterior $p(z|x)$. The hyperparamters can also be optimized, the detailed procedures for which are explained in the supplementary material.

## 4 Experiments

### 4.1 Experiments on synthetic datasets

#### 4.1.1 Data generation

We generated 10 datasets from PYP mixture models. Each dataset was generated as follows. We first generated cluster labels for 2,000 data points from $\mathrm{PYP}(\theta, \alpha)$ with $\theta = 1$ and $\alpha = 0.7$. Given the cluster labels, we generated data points from $\mathrm{Mult}(M, \omega)$, where the number of trials $M$ was chosen uniformly from $[1, 50]$ and $\omega$ was sampled from $\mathrm{Dir}(0.05 \cdot \mathbb{1}_{200})$. We also generated another 10 datasets from CRP mixture models $\mathrm{CRP}(\theta)$ with $\theta = 1$, to see if FNSPM adapts to the change of the underlying random measure. For each dataset, we used 80% of data points for training and the remaining 20% for testing.

#### 4.1.2 Algorithm settings and performance measure

We compared six algorithms - Collapsed Gibbs (CG) for FDPM (CG/D), Variational Bayes (VB) for FDPM (VB/D), Collapsed Variational Bayes (CVB) for FDPM (CVB/D), CG for FNSPM (CG/S),

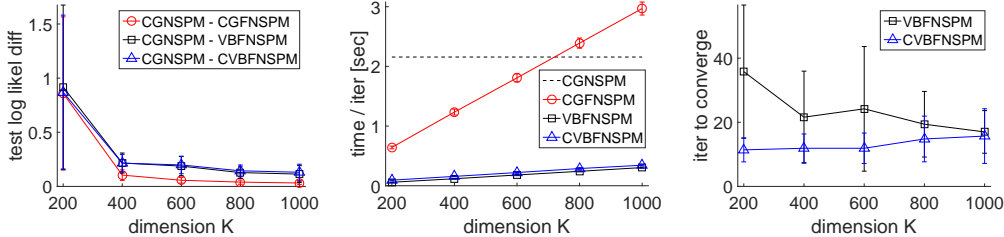

Figure 2: (Left) comparison between the infinite-dimensional algorithm and the finite dimensional algorithms. (Middle) Average times per iteration of the infinite and the finite dimensional algorithms. (Right) Average number of iterations need to converge for variational algorithms.

Table 1: Comparison between six finite-dimensional algorithms on synthetic PYP, synthetic CRP, AP corpus and NIPS corpus. Average test log-likelihood values and $\alpha$ estimates are shown with standard deviations.

| | PYP | | CRP | | AP | | NIPS | |
|---|---|---|---|---|---|---|---|---|
| | loglikel | $\alpha$ | loglikel | $\alpha$ | loglikel | $\alpha$ | loglikel | $\alpha$ |
| CG/D | -33.2078 (1.5557) | - | **-25.4076** (1.9081) | - | -157.2228 (0.0189) | - | -352.8909 (0.0070) | - |
| VB/D | -33.4480 (1.6495) | - | -25.4148 (1.9120) | - | -157.2379 (0.0304) | - | -352.9104 (0.0172) | - |
| CVB/D | -33.4278 (1.6525) | - | -25.4150 (1.9115) | - | -157.2302 (0.0280) | - | -352.8692 (0.0321) | - |
| CG/S | **-33.1039** (1.5676) | 0.6940 (0.0235) | -25.4079 (1.9077) | 0.2867 (0.0762) | -157.1920 (0.0036) | 0.5261 (0.0032) | -352.7487 (0.0037) | 0.5857 (0.0032) |
| VB/S | -33.1861 (1.5873) | 0.4640 (0.0085) | -25.5076 (1.9122) | 0.4770 (0.0041) | **-157.1391** (0.1154) | 0.4748 (0.0434) | **-352.6078** (0.2599) | 0.4945 (0.0324) |
| CVB/S | -33.2031 (1.5858) | 0.7041 (0.0322) | -25.4080 (1.9085) | 0.2925 (0.0608) | -157.2182 (0.0282) | 0.5327 (0.0060) | -352.7544 (0.0088) | 0.5899 (0.0070) |

VB for FNSPM (VB/S) and CVB for FNSPM (CVB/S). All the algorithms were initialized with a single run of sequential collapsed Gibbs sampling starting from zero clusters, and afterwards ran for 100 iterations. The variational algorithms were terminated if the improvements of the ELBO were smaller than a threshold. The hyperparameters $\theta$ and $\alpha$ were initialized as $\theta = 1$ and $\alpha = 0.5$ for all algorithms. The performances were measured by average test log-likelihood,

$$\frac{1}{N_{\text{test}}} \sum_{n=1}^{N_{\text{test}}} \log p(x_n | x_{\text{train}}). \tag{20}$$

For CG, we computed the average of samples collected every 10 iterations. For VB and CVB, we computed the log-likelihood using the expectations of the variational distributions.

### 4.1.3 Effect of $K$ on predictive performance and running time

To see the effect of $K$ on predictive performance, we first compared the finite-dimensional algorithms (CG for FNSPM, VB for FNSPM and CVB for FNSPM) to the infinite-dimensional algorithm (CG for NSPM [28]). We tested the four algorithms on 10 synthetic datasets generated from PYP mixtures, with $K \in \{200, 400, 600, 800, 1000\}$ for finite algorithms, and measured the difference of average test log likelihood compared to the infinite-dimensional algorithm. We also measured the average running time per iteration of the four algorithms, and the average number of iterations to converge of the variational algorithms. Figure 2 shows the results. As expected, the difference between finite-dimensional algorithms and the infinite-dimensional algorithm decreases as $K$ grows. The finite-dimensional algorithms have $O(NK)$ time complexity per iteration, and the infinite-dimensional algorithm has $O(N\tilde{K})$ where $\tilde{K}$ is the maximum number of clusters created during clustering. However, in practice, variational algorithms can be implemented with efficient matrix multiplications, and this makes them much faster than sampling algorithms. Moreover, as shown in Figure 2, variational algorithms usually converge in 50 iterations.

#### 4.1.4 Comparing finite-dimensional algorithms on PYP and CRP datasets

We compared six algorithms for finite mixture models (CG/D, VB/D, CVB/D, CG/S, VB/S and CVB/S) on PYP mixture datasets and CRP mixture datasets, with $K = 1000$. The results are summarized in Table 1. On PYP datasets, in general, FNSPM outperformed FDPM and CG outperformed VB and CVB. CG/S consistently outperformed CG/D, and the same relationship applied to VB/S and VB/D and CVB/S and CVB/D. Even though VB/S and CVB/S were variational algorithms, the performance gap between them and CG/S was not significant. Table 1 shows the estimated $\alpha$ values for CG/S, VB/S and CVB/S. CG/S and CVB/S seemed to recover the true value $\alpha = 0.7$, but VB/S didn't. We found that VB/S tends to control the other parameter $\theta$ while holding $\alpha$ near its initial value 0.5. On CRP datasets, all the algorithms showed similar performances except for VB/S, which was consistently worse than other algorithms. This is probably due to the bad estimates of $\alpha$.

#### 4.2 Experiments on real-world documents

We compared the six algorithms on real-world document clustering task by clustering AP corpus [1] and NIPS corpus [2]. We preprocessed the corpora using latent Dirichlet allocation (LDA) [3]. We ran LDA with 300 topics, and then gave each document a bag-of-words representation of topic assignments to those 300 topics. We assumed that those representations were generated from the multinomial-Dirichlet model, and clustered them using FDPM and FNSPM. We used 80% of documents for training and the remaining 20% for computing average test log-likelihood. We set $K = 2,000$ and ran each algorithm for 200 iterations. We repeated this 10 times to measure the average performance.The results are summarized in Table 1. In general, the algorithms based on FNSPM showed better performance than those of FDPM based ones, implying that FNSPM based algorithms are well capturing the heavy-tailed cluster distributions of the corpora. VB/S performed the best, even though it sometimes converged to poor values.

## 5 Conclusion

In this paper, we proposed finite-dimensional BFRY processes that converge to SP, GGP and SBP. The jumps of the finite-dimensional BFRY processes have nice closed-form densities, and this leads to the efficient posterior inference algorithms. With finite-dimensional BFRY processes, we developed variational Bayes and collapsed variational Bayes for finite-dimensional normalized SP mixture models, and demonstrated its performance both on synthetic and real-world datasets. As mentioned earlier, with finite dimensional BFRY processes one can develop variational Bayes or other posterior inference algorithms for a variety of models with SP, GGP and SBP priors. This fact, along with more theoretical properties of finite-dimensional processes, presents interesting avenues for future research.

**Acknowledgements**: This work was supported by IITP (No. B0101-16-0307, Basic Software Research in Human-Level Lifelong Machine Learning (Machine Learning Center)) and by National Research Foundation (NRF) of Korea (NRF-2013R1A2A2A01067464), and supported in part by the grant RGC-HKUST 601712 of the HKSAR.

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
