[Supplementary Material · 1574_supple.pdf]

# Supplementary Material for Finite-Dimensional BFRY Priors and Variational Bayesian Inference for Power Law Models

**Juho Lee**
POSTECH, Korea
stonecold@postech.ac.kr

**Lancelot F. James**
HKUST, Hong Kong
lancelot@ust.hk

**Seungjin Choi**
POSTECH, Korea
seungjin@postech.ac.kr

## 1 Proof of the convergence for finite-dimensional Gamma process

**Theorem 3.** *Let $\mu \sim \mathrm{GP}(\theta, H)$. Construct $\mu_K$ as follows:*

$$G_1, \ldots, G_K \overset{\text{i.i.d.}}{\sim} \mathrm{Gamma}(\theta/K, 1), \quad V_1, \ldots, V_K \overset{\text{i.i.d.}}{\sim} H, \quad \mu_K = \sum_{k=1}^{K} G_k \delta_{V_k}. \tag{1}$$

*Then, for arbitrary measurable $f$, $\lim_{K \to \infty} \mathcal{L}_f(\mu_K) = \mathcal{L}_f(\mu)$.*

*Proof.* The Laplace functional of $\mu_K$ is computed as:

$$
\begin{aligned}
\mathcal{L}_f(\mu_K) &= \left[ \int_0^\infty \int_\Omega \frac{s^{\theta/K-1} e^{-s}}{\Gamma(\theta/K)} e^{-f(\omega)s} ds H(d\omega) \right]^K \\
&= \left[ \int_\Omega (1 + f(\omega))^{-\theta/K} H(d\omega) \right]^K \\
&= \left[ 1 - \int_\Omega \{ 1 - (1 + f(\omega))^{-\theta/K} \} H(d\omega) \right]^K.
\end{aligned}
$$

By the Taylor's expansion of $1 - e^{-x}$ around 0, we have

$$
\begin{aligned}
1 - (1 + f(\omega))^{-\theta/K} &= 1 - \exp\left\{ -\frac{\theta}{K} \log(1 + f(\omega)) \right\} \\
&= \frac{\theta}{K} \log(1 + f(\omega)) - \frac{\theta^2}{2!K^2} \log(1 + f(\omega))^2 + \frac{\theta^3}{3!K^3} \log(1 + f(\omega))^3 + \ldots \\
&= \frac{\theta}{K} \Big[ \log(1 + f(\omega)) + O(K^{-1}) \Big].
\end{aligned}
$$

Hence, we have

$$\mathcal{L}_f(\mu_K) = \left[ 1 - \frac{1}{K} \int_\Omega \theta \Big[ \log(1 + f(\omega)) + O(K^{-1}) \Big] H(d\omega) \right]^K.$$

Since $O(K^{-1})$ is bounded for every $K$, by the bounded convergence theorem,

$$
\begin{aligned}
\lim_{K \to \infty} \mathcal{L}_f(\mu_K) &= \exp\left\{ -\int_\Omega \theta \log(1 + f(\omega)) H(d\omega) \right\} \\
&= \exp\left\{ -\int_0^\infty \int_\Omega (1 - e^{-sf(\omega)}) \theta s^{-1} e^{-s} ds H(d\omega) \right\},
\end{aligned}
$$

which is precisely $\mathcal{L}_f(\mu)$. $\qquad\square$

## 2 Variational Bayes for FNSPM

The density of the joint likelihood distribution of the FNSPM, with auxiliary variable $U$, is written as

$$p(x, z, u, s, \omega) = \frac{u^{N-1}}{\Gamma(N)} \prod_{k=1}^{K} \frac{\theta s_k^{N_k - \alpha - 1} e^{-us_k}(1 - e^{-(\alpha K/\theta)^{1/\alpha} s_k})}{K\Gamma(1-\alpha)} \left[ \prod_{z_n = k} \ell(x_n | \omega_k) \right] h(\omega_k), \quad (2)$$

where $\ell$ is the density for $L$ and $h$ is the density for $H$. We assume that $L(x|\omega)$ and $H(\omega_k)$ are conjugate exponential families:

$$\ell(x|\omega) = \exp\{\langle t(x), \omega \rangle - \langle \mathbb{1}, g(\omega) \rangle - b(x)\} \quad (3)$$
$$h(\omega) = \exp\{\langle \zeta, \omega \rangle - \langle \kappa, g(\omega) \rangle - c(\zeta, \kappa)\}, \quad (4)$$

and thus we have

$$\log p(x, z, u, s, \omega) = (N-1)\log u - \log \Gamma(N) + K(\log \theta - \log K - \log \Gamma(1-\alpha))$$
$$+ \sum_{k=1}^{K} \left[ (N_k - \alpha - 1)\log s_k - us_k + \log(1 - e^{-(\alpha K/\theta)^{1/\alpha} s_k}) \right.$$
$$\left. + \left\langle \zeta + \sum_{z_n=k} t(x_n), \omega_k \right\rangle - \langle \kappa + N_k \mathbb{1}, g(\omega_k) \rangle - \sum_{z_n=k} b(x_n) - c(\zeta, \kappa) \right]. \quad (5)$$

Now we introduce variational distributions for $\{u, s, z, \omega\}$, with density.

$$q(u, s, z, \omega) = q(u) \prod_{k=1}^{K} q(s_k) q(\omega_k) \prod_{n=1}^{N} \prod_{k=1}^{K} q(z_n = k). \quad (6)$$

The standard choice for $q(s_k)$ would be gamma distributions, but we found that the computing the expectation $\mathbb{E}_q[\log(1 - e^{-(\alpha K/\theta)^{1/\alpha} s_k})]$ is intractable, and numerical approximation methods are unstable especially for large $K$ values. Hence, we chose to do point estimations for $s_k$ by setting

$$q(s_k) = \delta_{\hat{s}_k}(s_k). \quad (7)$$

By doing this we cannot estimate the posterior variances of $s_k$, but we found that the point-estimates are enough for the model fitting and prediction in FNSPM. Finding good variational distributions for $q(s_k)$ would be a good future research direction. For the other variables, we assume that

$$q(u) = \delta_{\hat{u}}(u) \quad (8)$$
$$q(z_n = k) = \hat{r}_{nk} \quad (9)$$
$$q(\omega_k) = \exp\{\langle \hat{\zeta}_k, \omega_k \rangle - \langle \hat{\kappa}_k, g(\omega_k) \rangle - c(\hat{\zeta}_k, \hat{\kappa}_k)\}. \quad (10)$$

The variational parameters to optimize is then $\{\hat{u}, \{\hat{s}_k\}, \{\hat{r}_{nk}\}, \{\hat{\zeta}_k, \hat{\kappa}_k\}\}$. We optimize these parameters by maximizing the evidence lower bound (ELBO),

$$\mathcal{L} = \mathbb{E}_q[\log p(x, z, u, s, \omega)] - \mathbb{E}_q[\log q(z, u, s, \omega)]$$
$$= (N-1)\log \hat{u} - \log \Gamma(N) + K(\log \theta - \log K - \log \Gamma(1-\alpha))$$
$$+ \sum_{k=1}^{K} \left[ (\hat{N}_k - \alpha - 1)\log \hat{s}_k - \hat{u}\hat{s}_k + \log(1 - e^{-(\alpha K/\theta)^{1/\alpha} \hat{s}_k}) \right.$$
$$+ \left\langle \zeta + \sum_{n=1}^{N} \hat{r}_{nk} t(x_n) - \hat{\zeta}_k, \mathbb{E}[\omega_k] \right\rangle - \langle \kappa + \hat{N}_k \mathbb{1} - \hat{\kappa}_k, \mathbb{E}[g(\omega_k)] \rangle$$
$$\left. - \sum_{n=1}^{N} \hat{r}_{nk} b(x_n) + c(\hat{\zeta}_k, \hat{\kappa}_k) - c(\zeta, \kappa) \right] - \sum_{n=1}^{N} \sum_{k=1}^{K} \hat{r}_{nk} \log \hat{r}_{nk}, \quad (11)$$

where $\hat{N}_k := \sum_{n=1}^{N} \hat{r}_{nk}$.

**Update for $\hat{r}_{nk}$:** the gradient of $\mathcal{L}$ w.r.t. $\hat{r}_{nk}$ is given as

$$\frac{\partial \mathcal{L}}{\partial \hat{r}_{nk}} = \log \hat{s}_k + \langle t(x_n), \mathbb{E}[\omega_k] \rangle - \langle \mathbb{1}, \mathbb{E}[g(\omega_k)] \rangle + \text{const.}$$

By the properties of conjugate exponential families,

$$\mathbb{E}[\omega_k] = \frac{\partial c(\hat{\zeta}_k, \hat{\kappa}_k)}{\partial \hat{\zeta}_k}, \quad \mathbb{E}[g(\omega_k)] = -\frac{\partial c(\hat{\zeta}_k, \hat{\kappa}_k)}{\partial \hat{\kappa}_k}.$$

Hence,

$$\hat{r}_{nk} \propto \exp\left\{\log \hat{s}_k + \left\langle t(x_n), \frac{\partial c(\hat{\zeta}_k, \hat{\kappa}_k)}{\partial \hat{\zeta}_k}\right\rangle + \left\langle \mathbb{1}, \frac{\partial c(\hat{\zeta}_k, \hat{\kappa}_k)}{\partial \hat{\kappa}_k}\right\rangle\right\}. \tag{12}$$

**Update for $\{\hat{\zeta}_k, \hat{\kappa}_k\}$:** equating the gradients of $\mathcal{L}$ w.r.t. $\hat{\zeta}_k$ and $\hat{\kappa}_k$ gives

$$\hat{\zeta}_k = \zeta + \sum_{n=1}^{N} \hat{r}_{nk} t(x_n), \quad \hat{\kappa}_k = \kappa + \hat{N}_k \mathbb{1}. \tag{13}$$

**Update for $\hat{u}$:** equating the gradients of $\mathcal{L}$ w.r.t. $\hat{u}$ gives

$$\hat{u} = \frac{1}{N-1}\sum_{k=1}^{K} \hat{s}_k. \tag{14}$$

**Update for $\hat{s}_k$:** we first write the part of ELBO relevant to $\hat{s}_k$ as $f(s_k)$, as follows:

$$f(s_k) := (\hat{N}_k - \alpha - 1)\log \hat{s}_k - \hat{u}\hat{s}_k + \log(1 - e^{-\Phi \hat{s}_k}), \tag{15}$$

where $\Phi := (\alpha K/\theta)^{1/\alpha}$. Note that

$$\lim_{\hat{s}_k \to 0} f(s_k) = \lim_{\hat{s}_k \to 0} \log \hat{s}_k \left[(\hat{N}_k - \alpha - 1) + \frac{\log(1 - e^{-\Phi \hat{s}_k})}{\log \hat{s}_k}\right] = \lim_{\hat{s}_k \to 0} (\hat{N}_k - \alpha)\log \hat{s}_k, \tag{16}$$

since

$$\lim_{\hat{s}_k \to 0} \frac{\log(1 - e^{-\Phi \hat{s}_k})}{\log \hat{s}_k} = \lim_{\hat{s}_k \to 0} \frac{\Phi e^{-\Phi \hat{s}_k}/(1 - e^{-\Phi \hat{s}_k})}{1/\hat{s}_k} = \lim_{\hat{s}_k \to 0} \frac{\Phi \hat{s}_k}{e^{\Phi \hat{s}_k} - 1} = \lim_{\hat{s}_k \to 0} \frac{\Phi}{\Phi e^{\Phi \hat{s}_k}} = 1,$$

by L'Hospital's rule. Hence, when $\hat{N}_k < \alpha$, the maximization problem becomes ill-posed since it diverges to $\infty$ as $\hat{s}_k \to 0$. In such case, we set $\hat{s}_k$ to be the posterior expectation $\mathbb{E}[\hat{s}_k|\hat{u}]$, which is computed as

$$\mathbb{E}[\hat{s}_k|\hat{u}] = \int_0^{\infty} \hat{s}_k p(d\hat{s}_k|\hat{u}) = \frac{\alpha(1 - \xi^{1-\alpha})}{\hat{u}(\xi^{-\alpha} - 1)}, \tag{17}$$

where

$$p(\hat{s}_k|\hat{u}) = \frac{\alpha \hat{s}_k^{-\alpha-1} e^{-\hat{u}\hat{s}_k}(1 - e^{-\Phi \hat{s}_k})}{\Gamma(1-\alpha)\{(\hat{u}+\Phi)^\alpha - \hat{u}^\alpha\}}, \quad \xi := \frac{\hat{u}}{\hat{u}+\Phi}. \tag{18}$$

Otherwise, if $\hat{N}_k > \alpha$, we compute the gradient of $\mathcal{L}$ w.r.t. $\hat{s}_k$,

$$\frac{\partial \mathcal{L}}{\partial \hat{s}_k} = \frac{\hat{N}_k - \alpha - 1}{\hat{s}_k} - \hat{u} + \frac{\Phi e^{-\Phi \hat{s}_k}}{1 - e^{-\Phi \hat{s}_k}}. \tag{19}$$

We found that considering the geometry of the solution space is crucial in this maximization problem. Following [1], we update

$$\hat{s}_k \leftarrow \left|\hat{s}_k + \lambda_t \hat{s}_k \frac{\partial \mathcal{L}}{\partial \hat{s}_k}\right|, \tag{20}$$

where $\lambda_t$ is a learning rate. Here we applied the expended-mean representation updates in [1]. Note that taking the absolute value after update is equivalent to computing and updating the gradients of $\hat{s}_k = |\hat{s}_k'|$. We chose $\lambda_t = 0.05 \times (t+1)^{-0.5}$ for all experiments.

**Optimizing the hyperparameters:** we can optimize the hyperparemeters $\theta, \alpha$ by maximizing the ELBO. However, we found that the gradient descent or Newton's method work poorly in this case,

possibly because the objective functions have multiple modes. Instead, we chose to optimize $\theta$ and $\alpha$ via slice-sampling [2]; we sampled $\theta$ and $\alpha$ via a single iteration of slice sampling, and accepted those samples only if they improved the ELBO.

To sample $\theta$, we assumed a prior distribution $\theta \sim \text{Gamma}(a_\theta, b_\theta)$, and let $\theta = e^c$. Then, the log of the posterior density is given as

$$\log p(c|\dots) = (a_\theta + K)c - b_\theta e^c + \sum_{k=1}^{K} \log(1 - e^{-\Phi\hat{s}_k}) + \text{const.} \tag{21}$$

We sampled the new $v$ with this log density via slice sampling, and accepted it when the ELBO was improved. Likewise, to sample $\alpha$, we assumed the prior $\alpha \sim \text{Beta}(a_\alpha, b_\alpha)$ and let $\alpha = \frac{1}{1+e^{-r}}$. The log density is then

$$\log p(r|\dots) = -K \log \Gamma(1 - \alpha) + a_\alpha \log \alpha + b_\alpha \log(1 - \alpha) \tag{22}$$

$$+ \sum_{k=1}^{K} \Big[ -\alpha \log \hat{s}_k + \log(1 - e^{-\Phi\hat{s}_k}) \Big] + \text{const.} \tag{23}$$

We sampled new $r$ via slice sampling and accepted new values when the ELBO was improved.

## 3 Collapsed Gibbs sampler for FNSPM

By marginalizing the jumps $\{s_k\}$ from the joint likelihood, we get the following collapsed joint likelihood:

$$p(x, z, u, \omega) = \frac{u^{N-1}}{K^K \Gamma(N)} \prod_{k=1}^{K} \left[ \frac{\theta(1 - \xi^{N_k - \alpha})}{u^{N_k - \alpha}} \frac{\Gamma(N_k - \alpha)}{\Gamma(1 - \alpha)} \right]^{\mathbb{I}_{\{N_k > \alpha\}}} \left[ \frac{\theta u^\alpha}{\alpha} (\xi^{-\alpha} - 1) \right]^{\mathbb{I}_{\{N_k < \alpha\}}}$$

$$\times \left[ \prod_{z_n = k} \ell(x_n | \omega_k) \right] h(\omega_k). \tag{24}$$

We first marginalize out $\omega_k$ to get

$$p(x, z, u) = \frac{u^{K\alpha - 1} \theta^K}{K^K \Gamma(N)} \prod_{k=1}^{K} \left[ \frac{\Gamma(N_k - \alpha)}{\Gamma(1 - \alpha)} (1 - \xi^{N_k - \alpha}) \right]^{\mathbb{I}_{\{N_K > \alpha\}}} \left[ \frac{\xi^{-\alpha} - 1}{\alpha} \right]^{\mathbb{I}_{\{N_k < \alpha\}}} m(x_k), \tag{25}$$

where

$$m(x_k) = \int_\Omega \left[ \prod_{z_n = k} \ell(x_n | \omega) \right] h(\omega) d\omega$$

$$= \exp\left\{ h\left( \zeta + \sum_{z_n = k} t(x_n), \kappa + N_k \mathbb{1} \right) - c(\zeta, \kappa) - \sum_{z_n = k} b(x_n) \right\}. \tag{26}$$

Hence, the conditional posterior of $z_n$ is given as

$$p(z_n = k|\dots) \propto \begin{cases} (N_k^{\neg n} - \alpha) \dfrac{1 - \xi^{N_k^{\neg n} + 1 - \alpha}}{1 - \xi^{N_k^{\neg n} - \alpha}} \dfrac{m(x_n, x_k^{\neg n})}{m(x_k^{\neg n})} & \text{if } N_k^{\neg n} > \alpha \\[4mm] \dfrac{\alpha(1 - \xi^{1-\alpha})}{\xi^{-\alpha} - 1} \displaystyle\int_\Omega \ell(x_n | \omega) H(d\omega) & \text{if } N_k^{\neg n} < \alpha \end{cases}, \tag{27}$$

where the superscript $\neg n$ means except for the index $n$.

We use slice sampling to update $u$, $\theta$ and $\alpha$. The log density for $u = e^v$ is given as

$$\log p(v|\dots) = K\alpha v + \sum_{k=1}^{K} \left[ \mathbb{I}_{\{N_k > \alpha\}} \log(1 - \xi^{N_k - \alpha}) + \mathbb{I}_{\{N_k < \alpha\}} \log(\xi^{-\alpha} - 1) \right] + \text{const.} \tag{28}$$

The log density for $\theta = e^c$ is (with prior $\theta \sim \text{Gamma}(a_\theta, b_\theta)$)

$$\log p(c|\dots) = (K + a_\theta)c - b_\theta c$$

$$+ \sum_{k=1}^{K} \left[ \mathbb{I}_{\{N_k > \alpha\}} \log(1 - \xi^{N_k - \alpha}) + \mathbb{I}_{\{N_k < \alpha\}} \log(\xi^{-\alpha} - 1) \right] + \text{const.} \tag{29}$$

The log density for $\alpha = \frac{1}{1 + e^{-r}}$ (with prior $\alpha \sim \text{Beta}(a_\alpha, b_\alpha)$) is

$$\log p(r|\dots) = (K\alpha \log u + a_\alpha \log \alpha + b_\alpha \log(1 - \alpha)$$

$$+ \sum_{k=1}^{K} \left[ \mathbb{I}_{\{N_k > \alpha\}} (\log \Gamma(N_k - \alpha) - \log \Gamma(1 - \alpha) + \log(1 - \xi^{N_k - \alpha})) \right.$$

$$\left. + \mathbb{I}_{\{N_k < \alpha\}} (\log(\xi^{-\alpha} - 1) - \log \alpha) \right] + \text{const.} \tag{30}$$

## 4  Collapsed Variational Bayes for FNSPM

Based on the collpased joint likelihood (24), we can develop a collapsed variational Bayes for TNSPM. We introduce variational distributions for $\{z, u, \omega\}$ as

$$q(z, u, \omega) = q(u) \prod_{k=1}^{K} q(\omega_k) \prod_{n=1}^{N} \prod_{k=1}^{K} q(z_n = k), \tag{31}$$

where

$$q(u) = \delta_{\hat{u}}(u) \tag{32}$$
$$q(z_n = k) = \hat{r}_{nk} \tag{33}$$
$$q(\omega_k) = \exp\{\langle \hat{\zeta}_k, \omega_k \rangle - \langle \hat{\kappa}_k, g(\omega_k) \rangle - h(\hat{\zeta}_k, \hat{\kappa}_k)\}. \tag{34}$$

**Update for $\hat{r}_{nk}$:** the update equation for $q(z_n = k)$ is given as

$$q(z_n = k) \propto \exp \left\{ \mathbb{E}_q[\log p(z_n | z^{\neg n})] + \left\langle t(x_n), \frac{\partial c(\hat{\zeta}_k, \hat{\kappa}_k)}{\partial \hat{\zeta}_k} \right\rangle + \left\langle \mathbb{1}, \frac{\partial c(\hat{\zeta}_k, \hat{\kappa}_k)}{\partial \hat{\kappa}_k} \right\rangle \right\}. \tag{35}$$

Following [3], we use the Gaussian approximation to compute $\mathbb{E}_q[\log p(z_n | z^{\neg \neg n})]$. For FNSPM, the conditional distribution is

$$p(z_n = k | z^{\neg n}) \propto \begin{cases} (N_k^{\neg n} - \alpha) \dfrac{1 - \xi^{N_k^{\neg n} + 1 - \alpha}}{1 - \xi^{N_k^{\neg n} - \alpha}} & \text{if } N_k^{\neg n} > \alpha \\[3mm] \dfrac{\alpha(1 - \xi^{1-\alpha})}{\xi^{-\alpha} - 1} & \text{if } N_k^{\neg n} < \alpha \end{cases}, \tag{36}$$

For convenience, let $f(N_k^{\neg n}) = \log p(z_n = k | z^{\neg n})$. Then by the second order Taylor's approximation we have

$$\mathbb{E}[f(N_k^{\neg n})] \approx f(\mathbb{E}[N_k^{\neg n}]) + \frac{1}{2} f''(\mathbb{E}[N_k^{\neg n}]) \text{Var}(N_k^{\neg n}). \tag{37}$$

In practice, we found that the first order approximation is enough:

$$\mathbb{E}[f(N_k^{\neg n})] \approx f(\mathbb{E}[N_k^{\neg n}])$$

$$= \mathbb{E}[\mathbb{I}_{\{N_k^{\neg n} > \alpha\}}] \left\{ \log(\hat{N}_k^{\neg n} - \alpha) + \log(1 - \xi^{\hat{N}_k^{\neg n} + 1 - \alpha}) - \log(1 - \xi^{\hat{N}_k^{\neg n} - \alpha}) \right\}$$

$$+ \mathbb{E}[\mathbb{I}_{\{N_k^{\neg n} < \alpha\}}] \left\{ \log \alpha + \log(1 - \xi^{1-\alpha}) - \log(\xi^{-\alpha} - 1) \right\}, \tag{38}$$

where $\hat{N}_k^{\neg n} = \mathbb{E}_q[N_k^{\neg n}] = \sum_{n' \neq n} \hat{r}_{n'k}$. We further approximate this as

$$\mathbb{E}[f(N_k^{\neg n})] \approx \begin{cases} \log(\hat{N}_k^{\neg n} - \alpha) + \log(1 - \xi^{\hat{N}_k^{\neg n} + 1 - \alpha}) - \log(1 - \xi^{\hat{N}_k^{\neg n} - \alpha}) & \text{if } \hat{N}_k^{\neg n} > \alpha \\[3mm] \log \alpha + \log(1 - \xi^{1-\alpha}) - \log(\xi^{-\alpha} - 1) & \text{if } \hat{N}_k^{\neg n} < \alpha \end{cases}, \tag{39}$$

**Update for** $\hat{u}$: as in the variational Bayes, we found that the gradient descent works poorly. Hence, we use slice-sampling using the log density (28) and accepted samples only if the ELBO was improved.

**Optimizing the hyperparameters**: again we use slice-sampling with log densities (30), (29) and accepted samples samples only if the ELBO was improved.