[Reviews · NeurIPS 2016]

Reviewer 1

Summary

This paper considers finite-dimensional approximations to the stable, generalized gamma, and stable beta processes. The construction uses scaled and exponentially tilted versions of the BFRY distribution. The main advantage of this approximation, is that the random variables involved can be simulated easily and admit tractable probability density functions, which makes them amenable to the implementation of variational algorithms.

Qualitative Assessment

This paper considers finite-dimensional approximations to the stable, generalized gamma, and stable beta processes. The construction uses scaled and exponentially tilted versions of the BFRY distribution. The main advantage of this approximation, is that the random variables involved can be simulated easily and admit tractable probability density functions, which makes them amenable to the implementation of variational algorithms. The paper is well written and I find the contributions of the paper of interest and potentially useful. The main contributions of the papers are in section 3.2, where the authors show the weak convergence of the finite-dimensional approximations of the stable, generalized gamma dn stable beta processes, using Laplace functional. A weaker point of the paper is the experimental part, which is somewhat lacking and may be improved in order to fully demonstrate the usefulness of the approach. The experiments on synthetic data in Section 4.1 aim at (i) characterizing the effect of the parameter K, that controls the approximation error and (ii) compare different algorithms for fitting the finite-dimensional models. The authors use a Pitman-Yor process to simulate the data, then consider a normalized stable process, a Dirichlet process, and their finite-dimensional approximations for inference. I don't really understand this choice; in order to evaluate the effect of the approximation it would have been more suitable to simulate from the process whose approximation is taken for inference. (e.g. simulate from a normalized stable process and use the finite-dimensional approximation for inference). Also, I don't see why the authors only consider the normalized stable and Dirichlet process: the first process is not really used as a BNP prior, and VB algorithms already exist for the DP; it would have made more sense to use the generalized gamma process, which includes both processes as special cases, and estimate its parameters. Minor comments: * To me, the paper introduces far too many acronyms, and it becomes difficult to follow at some point * line 141: u^{\theta/K - 1} is not bounded by 1 * Section 3.1: a reference regarding the weak convergence of random measures might be useful, e.g. the book Daley and Vere-Jone (2008) * Section 3.1. and appendix: the proof of of weak convergence of a finite-dimensional gamma process already appears in Ishwaran and James (2004) (cited by the authors [5]). The proof in the appendix is therefore not necessary. A reference to this paper may be useful in section 3.1. * Line 161: This is valid for any value of tau not just tau=1. * Line 206: I dont' think the acronym PYP has been introduced. * Line 190,201: hyperparameters * Line 215: Collapsed * Line 206: ran until for * Line 224: dimensional

Confidence in this Review

2-Confident (read it all; understood it all reasonably well)


Reviewer 2

Summary

The paper describes a class of finite-dimensional processes that converge to certain completely random measures of interest, such as the stable process, the generalized Gamma process and the stable Beta process. These finite-dimensional processes have explicit densities and are easy to simulate, making it easy to derive variational inference and Gibbs sampling schemes for models based on these processes, thereby potentially making them applicable to larger datasets which cannot be handled by current complex MCMC schemes. The performance of three algorithms designed for these processes, applied to a normalized stable process mixture model is compared on synthetic data and validated on a document clustering task.

Qualitative Assessment

The paper is well-structured and written clearly (modulo some typos and missing articles). The contribution is relevant and potentially impactful, as scalable inference procedures for models based on the considered stochastic processes are currently not available, and the paper provides a stepping stone for the development of such algorithms (e.g. based on SVB). The contribution appears novel and references relevant related work. The empirical evaluation could be strengthened, especially regarding the use of the finite dimensional processes to scale up inference in the corresponding infinite-dimensional models. In particular, it would be nice to see the predictive performance vs. runtime of the finite-dimensional model at various truncation levels and the infinite-dimensional variant.

Confidence in this Review

2-Confident (read it all; understood it all reasonably well)


Reviewer 3

Summary

The authors propose a novel class of finite-dimensional BFRY processes constructed from iid BFRY random variables with positive mass on a finite number of atoms. They show that in the limit as the number of atoms approaches infinity, the finite-dimensional BFRY processes may converge to a Stable process (SP), Generalised Gamma process (GGP), or a Stable Beta process (SBP). Interest in processes such as the SP, GGP, and SBP process has been growing as they possess a power law behaviour; however, posterior inference can be difficult and expensive. Importantly, finite dimensional BFRY processes are easily simulated and possess a closed form for the density of the jumps. This makes them an ideal approximation to the infinite dimensional SP, GGP, or SBP as both MCMC algorithms and variational inference algorithms can be developed. The authors develop a variational Bayes, collapsed variational Bayes, and collapsed Gibbs sampling algorithm for the finite dimensional BFRY process mixture models for topic modelling in simulated and real examples.

Qualitative Assessment

The paper is interesting, well-written and well-motivated. My main concerns lie with Section 4 on the experiments: The description of the simulated data needs to be clarified. In generating data from Mult(M,\omega) state what is cluster specific or data point specific. For the model, please state what is the base measure and its parameters. Only 200 iterations is quite short for a Gibbs sampling algorithm, the authors should comment on this choice and if diagnostic tests showed convergence. Similarly, was convergence reached for the variational Bayes algorithms? I was quite surprised to see that VB outperformed CG and CVB in real examples, maybe this is due to lack of convergence? The paper could also be strengthen with a discussion on computational complexity of the algorithms, particularly in comparison with algorithms for the infinite dimensional process, as this was one of the main motivations of the paper. You are performing Bayesian inference on the parameters of the Stable process in the collapsed Gibbs sampling algorithm but are finding Empirical Bayes point estimates of the parameters in VB and CVB; what are the "estimates"of alpha plotted in Figures 2, 3, 4,5 for collapsed Gibbs, is this the posterior mean as a function of iteration or simply the samples (the posterior mean would be a more appropriate estimate)? please also explain how the confidence/credible bars are derived. Other main concerns: There appears to be an error in the proof of "Theorem 1" in the last line before equation 8; $f(\omega)$ has disappeared and this last step needs to be clarified. Comments: 1. pg. 2 line 58: processes that HAVE behaviour similar to... 2. pg. 3 line 93: Laplace functional is COMPUTED as... 3. pg. 3 line 106: The Stable Beta process.... 4. pg. 6 line 188: define $z=(z_1,....z_n)$, $s=(s_1,....s_k)$, and $\omega=(\omega_1,....\omega_k)$. 5. pg. 6 line 193: to get THE collapsed model. 6. pg. 6 line 201: The HYPERPARAMETERS can also... 7. pg. 7 line 215: COLLAPSED Variational Bayes 8. pg. 8 line 217: clarify what is meant by "started from no clusters": all data points in one cluster? or all data points in their own cluster? 9. pg. 8 line 261:"for any type of models with SP, GGP, and SBP priors" may be an overstatement. Supplementary material comments: 1. pg. 1 line 2: Please do not call this Theorem 1, as it is confusing with the Theorem 1 in main text. 2. pg. 2 eq. 6: variation distribution for $u$ is missing. 3. pg. 2 line 20: good future research DIRECTION. 4. pg. 4 line 47: transformation should be $\theta=\exp(c)$. 5. pg. 4 line 60: transformation should be $u=\exp(v)$. 6. pg. 5 line 61: transformation should be $\theta=\exp(c)$. 7. pg. 6 line 62: the Beta prior should be on \alpha not r, as \alpha takes values in the unit interval. 8. pg. 6 first eq after line 62: I believe should be (K\alpha-1)log(u), i.e. there is a missing -1 here.

Confidence in this Review

2-Confident (read it all; understood it all reasonably well)


Reviewer 4

Summary

Priors for random measures that induces power-law distributions (such as the GGP) has recently captured a great deal of attention in Bayesian non-parametrics. Inference in models based on these measures is often made difficult because of their analytical properties, for instance the difficulty in obtaining a simple representation of the posterior distribution. This makes existing samplers slow and difficult to implement and makes the use of other inference techniques such as VB difficult. The authors propose a finite-dimensional (truncated) representation of the underlying random measure based on the BFRY distribution which allows simpler sampling procedures and easy construction of VB inference. The construction is tested on realistic data sets.

Qualitative Assessment

This paper addresses a timely and important problem in modern non-parametrics, efficient inference for an important class of power-law inducing random measures. I was not aware of the BFRY distribution and pointing out tools such as this, and how they can be implemented, is in my view a good contributions to this conference. I must however admit I did not understand all details of the paper and I hope other reviewers will be able to confirm it's technical accuracy. I understand this is a difficult and technical topic however I think the authors should improve the presentation. Many difficult topics are introduced or mentioned very quickly and it would serve the paper well if any of these could be omitted for brevity. Is it necessary to introduce the sigma-stable random variable, Kingmann processes, and tilting in the *introduction*? A suggestion could be to introduce the representation eq. (14) in the introduction as an equation (perhaps with generic distributions instead of the BFRY) to provide the reader with a clear picture of what we want to end up with and what the concept of "ideal" finite-dimensional distribution refers to. If the authors need more space, eq.13-15 contains some redundancy. In regards to the simulations, I think the graphs needs to be cleaned up. In my printed version they are hardly readable and more thought should be given to their readability (font/line sizes, data area, etc.). Should the figure on p.4 not be named? I would also recommend giving the references size/spacing some more thought (are you sure this agrees with the NIPS format?). I would suggest not showing the difference in figure 1 but the absolute value. Later, the error bars overlap in a way that makes the graphic very difficult to read. I think the tables should be cleaned up to either exclude variance or show it in bolded; quite frankly I got lost in terms of what conclusions I should draw from the tables because of the many abbreviations and it would be recommendable to show a subset of the results and better communicate what the conclusions are. I would also suggest showing a single number and bolding “best” entries. A thing I missed a lot from the simulations are results showing that the power-law behavior of the finite-dimensional representation is easily preserved. Playing the devils advocate, the prior distribution (DP or something with a power low behavior) is sometimes not that important in data modelling, and therefore a good test log-likelihood in the actual models does not necessarily show the finite-dimensional representation preserves power-law behavior. It would in my view benefit the paper if it could be shown by simulation for which K (compared to some dataset size simulated from a power-law distribution such as a GGP) power-law behavior in the posterior is preserved by the finite-dimensional representation. One could expect the tail distribution to require a lot of simulated atoms (high K) to accurately preserve the tail and therefore the power-law somewhat limiting the utility of the result and disproving this would strengthen the paper in my view. Overall I think this is a good paper that deserves to be on NIPS provided some editorial effort is given to the presentation.

Confidence in this Review

2-Confident (read it all; understood it all reasonably well)


Reviewer 5

Summary

This paper introduces finite-dimensional processes that converge to completely random measures (CRM) as the dimension K goes to infinity. Unlike standard truncations of CRM, obtained by only considering the first K jumps as in the Ferguson & Klass algorithm eg, the proposed methodology enjoys fast simulation due to iid simulation of the jumps. Additionally, closed form expressions for the jump densities allow for VB type algorithm techniques.

Qualitative Assessment

I find this a thought-provoking paper as it relates the little known BFRY distribution to popular infinite dimensional processes that are the generalized gamma process (and stable process as a special case) and the stale beta process. Comments: - since it is (at least to me) little known, the reference associated to BFRY could be recalled earlier, eg in Introduction. - the main contribution of the paper is to introduce a finite dimensional approximation to infinite dimensional processes. There are a few related works that provide approximation sampling of CRMs that would be relevant citations here. [1] proposes an adaptive truncation sampler based on sequential Monte Carlo, while [2] builds upon the Ferguson & Klass algorithm. - related to the approximation, an actual assessment of the quality of approximation is proposed in section 4.1.4 and Fig 1. Related is the approximation assessment proposed by [2] based on comparing moments of the finite and infinite dimensional CRM, hence having the advantage to account also for unsampled jumps. - SBP on line 107: the assumption on c is that it is larger than -\alpha in the original paper by Teh & Gorur. Also, add "du" at the end of Eqn (6) - the finite dimensional version for the SBP is only obtained for the c=0 special case. Could you comment about the reason why? [1] Griffin, J. E. (2016). An adaptive truncation method for inference in bayesian nonparametric models. Statistics and Computing, 26(1-2):423–441. [2] Arbel, J. and Prünster, I. (2016). A moment-matching Ferguson & Klass algorithm. Statistics and Computing, in press.

Confidence in this Review

3-Expert (read the paper in detail, know the area, quite certain of my opinion)


Reviewer 6

Summary

Paper describes several finite dimensional processes with closed form densities for jumps, converging to some of the important infinite dimensional processes. Several inference algorithms for FNSPM based on the new representations are proposed.

Qualitative Assessment

Theorem 2 of the paper seemed novel and potentially promising to me. Unfortunately experimental results and proposed inference algorithms did not demonstrate usefulness of the representation provided in the theorem. Almost half of the exposition is giving some background information, instead I would be interested to see more results utilizing proposed finite dimensional representations of the various processes. In particular, if the suggested representations would lead to faster\more accurate inference in the nonparametric setting. I'm concerned about the design of the real-world experiment. LDA is a mixed membership clustering model finding topics and their proportions in documents. What do we learn by using DPM on top of the 'bag of topics' of the LDA? While abstract is posing this work as one leading to scalable inference, scalability is not really addressed in the paper. Tables are not easy to read, i.e. lines are too close to numbers.

Confidence in this Review

2-Confident (read it all; understood it all reasonably well)